# A high-performance capillary-fed electrolysis cell promises more cost-competitive renewable hydrogen

Aaron Hodges[1,3], Anh Linh Hoang[1,3], George Tsekouras[1], Klaudia Wagner[1,2], Chong-Yong Lee [1,2], Gerhard F. Swiegers [1,2 ✉] & Gordon G. Wallace [1,2]

Renewable, or *green*, hydrogen will play a critical role in the decarbonisation of hard-to-abate sectors and will therefore be important in limiting global warming. However, renewable hydrogen is not cost-competitive with fossil fuels, due to the moderate energy efficiency and high capital costs of traditional water electrolysers. Here a unique concept of water electrolysis is introduced, wherein water is supplied to hydrogen- and oxygen-evolving electrodes via capillary-induced transport along a porous inter-electrode separator, leading to inherently *bubble-free* operation at the electrodes. An alkaline *capillary-fed electrolysis* cell of this type demonstrates water electrolysis performance exceeding commercial electrolysis cells, with a cell voltage at 0.5 A cm$^{-2}$ and 85 °C of only 1.51 V, equating to 98% energy efficiency, with an energy consumption of 40.4 kWh/kg hydrogen (vs. ~47.5 kWh/kg in commercial electrolysis cells). High energy efficiency, combined with the promise of a simplified balance-of-plant, brings cost-competitive renewable hydrogen closer to reality.

---

[1] Intelligent Polymer Research Institute, University of Wollongong, Wollongong, NSW 2522, Australia. [2] Australian Research Council Centre of Excellence for Electromaterials Research, University of Wollongong, Wollongong, NSW 2522, Australia. [3] These authors contributed equally: Aaron Hodges, Anh Linh Hoang. ✉email: swiegers@uow.edu.au

A nthropogenic climate change, driven largely by the burning of fossil fuels, poses a global existential threat. This has motivated a growing number of nations and corporations to aim for net-zero carbon emissions by 2050 to limit global warming to 1.5 °C above pre-industrial levels[1,2].

A critical element of the future net-zero world will be renewable hydrogen, or green hydrogen, produced by water electrolysis powered by renewable electricity, such as solar and wind. The electrolysis of water requires the input of electrical energy and heat energy, resulting in the evolution, from water, of hydrogen gas at the cathode and oxygen gas at the anode according to:

$$2H_2O_{(l)} \rightleftharpoons 2H_{2(g)} + O_{2(g)} \quad E^0 = -1.229\,V$$

*Green* hydrogen will be essential to the decarbonisation of hard-to-abate sectors such as steel manufacture, long-haul transport, shipping and aviation[1–3]. It may also be used for the seasonal storage of renewable electricity[1–7] and as a chemical feedstock[1–5]. However, the levelised cost of green hydrogen (LCOH) is presently not competitive with fossil fuels. This is due to the high capital expenditure (CAPEX) and high operational expenditure (OPEX) of present-day water electrolysers. The OPEX is, by far, the larger component of LCOH and it is dominated by the overall energy efficiency of the water electrolyser and the cost of the input renewable electricity to which it applies[2]. At sub-MW scale, state-of-the-art commercial water electrolysers typically require ~53 kWh of electricity to produce 1 kg of hydrogen, which contains 39.4 kWh of energy, according to its higher heating value (HHV)[2]. Of that, the electrolysis cell, which is ~83% energy efficient (HHV) at the operating current density, consumes ~47.5 kWh, with the engineering system, known as the balance-of-plant, consuming the remaining ~5.5 kWh[2]. The International Renewable Energy Agency (IRENA) has set a 2050 target[2] to decrease cell energy consumption to <42 kWh/kg. Any improvements in net energy efficiency create a proportionally equivalent decrease in the levelised cost of the produced hydrogen (Supplementary Fig. 1).

This work introduces a unique concept of water electrolysis that promises notably reduced CAPEX and OPEX compared to conventional water electrolysers, making renewable hydrogen more cost-competitive with fossil fuels.

Inspired by the historic evolution of water electrolysis cells, which recently culminated in asymmetric polymer electrolyte membrane (PEM) cells that directly produce one of the gases in a gas collection chamber rather than bubbling through the liquid electrolyte[8], we have developed a capillary-fed electrolysis (CFE) cell concept in which both gases are produced directly in gas collection chambers (Fig. 1). The aqueous electrolyte is constantly supplied to the electrodes by a spontaneous capillary action in the porous, hydrophilic, inter-electrode separator. The bottom end of the separator is dipped in a reservoir, resulting in capillary-induced, upward, in-plane, movement of electrolyte. Porous gas diffusion electrodes are held against opposite sides of the separator, above the level of the electrolyte. The electrodes draw in liquid laterally from the separator and are covered with a thin layer of the electrolyte. The application of sufficient voltage between the electrodes results in the electrolysis of water, which is continuously replenished by water moving up the separator from the reservoir. Because the generated hydrogen and oxygen gases readily migrate through the thin layer of liquid electrolyte covering their respective electrodes, the capillary-fed cell concept provides for bubble-free electrolysis in which water is converted directly to the bulk gases without forming gas bubbles[9–13].

In so doing, the cell avoids bubbles masking the electrodes and maintains access to the catalytic sites on the electrodes. This includes access to the most active crevice, cleft and defect sites that are the first to be blocked by bubble formation, sometimes permanently so[14]. It also ensures that water flow to an electrode does not counteract gas flow away from the electrode, thereby avoiding the counter multi-phase flows inherent in conventional water electrolysers and their associated mass transport limitations. In diminishing the energy needed to overcome such inefficiencies, the capillary-fed cell realises significantly improved energy efficiency.

It is also important to assess whether new cell configurations increase or decrease the complexity (i.e. the energy consumption and CAPEX) of the balance-of-plant. In the case of CFE cells, notable simplifications in the balance-of-plant are apparent. The absence of gas bubbles and associated gas-liquid froth formation

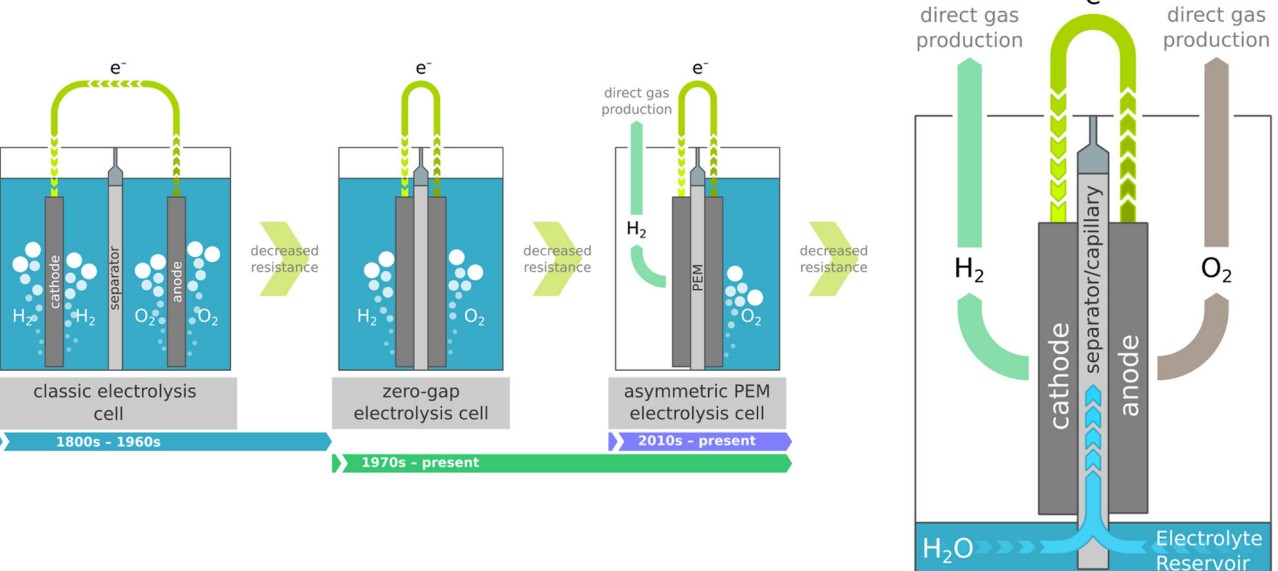

**Fig. 1 Conceptualisation of the Capillary-Fed Electrolysis (CFE) cell.** Inspired by the historic evolution of water electrolysis cell architectures culminating in the direct production of one of the gases, the Capillary-Fed Electrolysis cell directly produces both gases. Liquid electrolyte is continuously drawn up the separator by a capillary effect, from a reservoir at the bottom of the cell. The porous, hydrophilic separator sustains the flow rate required for water electrolysis.

in the cell stack, removes the need for liquid circulation, eliminating the gas-liquid separator tanks normally required and their piping, pumps, and fittings (Supplementary Figs. 2–3). The high energy efficiency, further, permits air-cooling, or radiative self-cooling, eliminating need for water-cooled chillers (Supplementary Tables 1–2). The small volumes of liquid electrolyte in each cell reservoir decrease the overall volume of water required (Supplementary Table 3). Unwanted and wasteful shunt currents found in conventional alkaline water electrolysers, may also be avoided. These simplifications in the balance-of-plant lead to downward pressure on electrolyser CAPEX.

The capillary-induced flow of aqueous 27 wt% KOH electrolyte up a saturated, hydrophilic, porous, polyether sulfone (PES) separator is initially measured and modelled, demonstrating its ability to indefinitely support water electrolysis at 1 A cm$^{-2}$ and ≥80 °C for a height of up to 18 cm. This height restriction, which is created by gravity, is taken into account here, although it may in future be avoided by locating the reservoir at the top of the separator.

In this work we show that a capillary-fed cell, employing a known NiFeOOH oxygen evolution electrocatalyst on the anode and Pt/C hydrogen evolution electrocatalyst on the cathode, tested at 80–85 °C with 27 wt% KOH electrolyte, yields water electrolysis with performance that exceeds conventional, bubbled control cells, and commercial alkaline and PEM cells. Faradaic efficiencies approach 100%, with low gas crossover. Cell energy efficiencies at 85 °C of 95% (HHV) at 0.8 A cm$^{-2}$ and 100% (HHV) at 0.3 A cm$^{-2}$ (39.4–41.6 kWh kg$^{-1}$ H$_2$) surpass the IRENA 2050 target and combine with the promise of a simplified balance-of-plant to bring cost-competitive renewable hydrogen closer to reality.

## Results

### Model for in-plane capillary-induced transport of a liquid through a porous material.

The starting point for modelling of in-plane transport of a liquid within a porous material under a capillary action, is the Hagen–Poiseuille equation, to which a term for tortuosity ($\tau$) was added to reflect the actual distance travelled by the liquid [Eq. 1]:

$$Q = \frac{n\pi r^4 \Delta P}{8\tau\mu L} \quad (1)$$

where $Q$ is the flow rate, $n$ is the number of capillaries of radius $r$, $\Delta P$ is the pressure drop, $\mu$ is the viscosity of the liquid, and $L$ is the length of the porous material.

Equations for porosity, and for the tortuosity of porous membranes[15], as well as the Young–Laplace equation, are shown below:

$$\varepsilon = \frac{n\pi r^2}{A} \quad (2)$$

$$\tau = \frac{(2-\varepsilon)^2}{\varepsilon} \quad (3)$$

$$\Delta P = \frac{2\gamma cos\theta}{r} \quad (4)$$

where $\varepsilon$ is porosity of the porous material, $A$ is the cross-sectional area of the porous material, $\gamma$ is the surface tension of the liquid, $\theta$ is the contact angle of the liquid on the material that comprises the porous material, and $r$ is the average pore radius of the porous material.

Substituting Eqs. (2), (3), and (4) into (1) yields:

$$Q = \frac{\varepsilon^2 A r \gamma cos\theta}{4(2-\varepsilon)^2 \mu L} \quad (5)$$

Eq. (5) permits a first-principles calculation of the capillary-induced rate of in-plane transport of a liquid (e.g. an aqueous

KOH electrolyte) through a thin porous material using measurable or known quantities.

### Capillary-induced, in-plane transport of aqueous KOH electrolyte within porous polyether sulfone (PES) filters.

A search was undertaken to identify potential inter-electrode separators that could draw up aqueous KOH electrolyte by a capillary action. This led to a series of commercially available porous, hydrophilic polyether sulfone filtration membranes that were specified as having average pore diameters of 0.45 μm, 1.2 μm, 5 μm, and 8 μm. Each was characterised for its capacity to draw up a 27 wt% aqueous KOH solution from a reservoir by capillary action. This electrolyte closely represents, at 20–80 °C, the 6 M KOH that has historically been used in industrial alkaline electrolysers. The capillary flow rates at different heights, when the filter was full of liquid, were measured (as described in the Method section and in Supplementary Fig. 4). A linear flow regime, termed Darcy flow, is observed. A dry filter filling itself for the first time exhibits non-linear Washburn flow (Supplementary Fig. 4).

Figure 2a shows the capillary-induced, in-plane, Darcy flow rates of the polyether sulfone filters at room temperature. Modelled flow rates, using Eq. (5), showed reasonable agreement with the measured flow rates (Source Data—Figs. 2–4). The contact angle (70.3°) was measured with the captive bubble technique; literature values provided surface tension and viscosity[16,17].

The flow rates can be seen to decline with increasing height and with smaller pores. The polyether sulfone filter having 8 μm average pore diameter demonstrated the highest flow rates (Fig. 2a(i)) and was selected for further use. If employed as an inter-electrode separator, this filter has a modelled flow rate at room temperature sufficient to indefinitely support capillary-fed water electrolysis at 0.5 A cm$^{-2}$ up to a height of at least 15 cm (lower dotted line in Fig. 2a(i)). Modelling using Eq. (5), indicated that at ≥80 °C, water electrolysis at 1 A cm$^{-2}$ could be supported up to at least 18 cm in height (upper dotted line in Fig. 2a(i)), permitting the construction of water electrolysis cells with practical height.

### Pore structure, porosity, and the ionic resistance of the polyether sulfone (PES) separator.

Further investigations of the polyether sulfone separator with 8 μm average pore diameter revealed it to have a porous, open structure. Figure 2b depicts SEM micrographs of its cross-section and two sides, one of which had a gloss, and the other a matte appearance. It was found to be 80% porous (see the Methods section).

The ionic resistance of the polyether sulfone filter, when filled with 27 wt% KOH, was measured as described in Supplementary Fig. 5. Table 1 compares its ionic resistance at room temperature with those of separators typically used in commercial alkaline (Zirfon PERL UTP 500)[18] and PEM electrolysis cells (Nafion$^{TM}$ 115 and Nafion$^{TM}$ 117)[19]. All of these ionic resistances include the resistance of electrolyte incorporated within them. The polyether sulfone separator displayed 244 mΩ cm$^2$ lower ionic resistance than Zirfon PERL UTP 500, due to its lesser thickness and higher porosity. Its ionic resistance was also 174 mΩ cm$^2$ less than Nafion$^{TM}$ 117 and 96 mΩ cm$^2$ less than Nafion$^{TM}$ 115. The low ionic resistance of the PES separator compared to commercial separators contributes to the high energy efficiency of the capillary-fed cell described below.

### Capillary-fed electrolysis cell outperforms conventional and commercial water electrolysis cells.

To evaluate the CFE cell concept, a test cell was fabricated using the polyether sulfone filter with 8 μm average pore diameter, as the electrode separator.

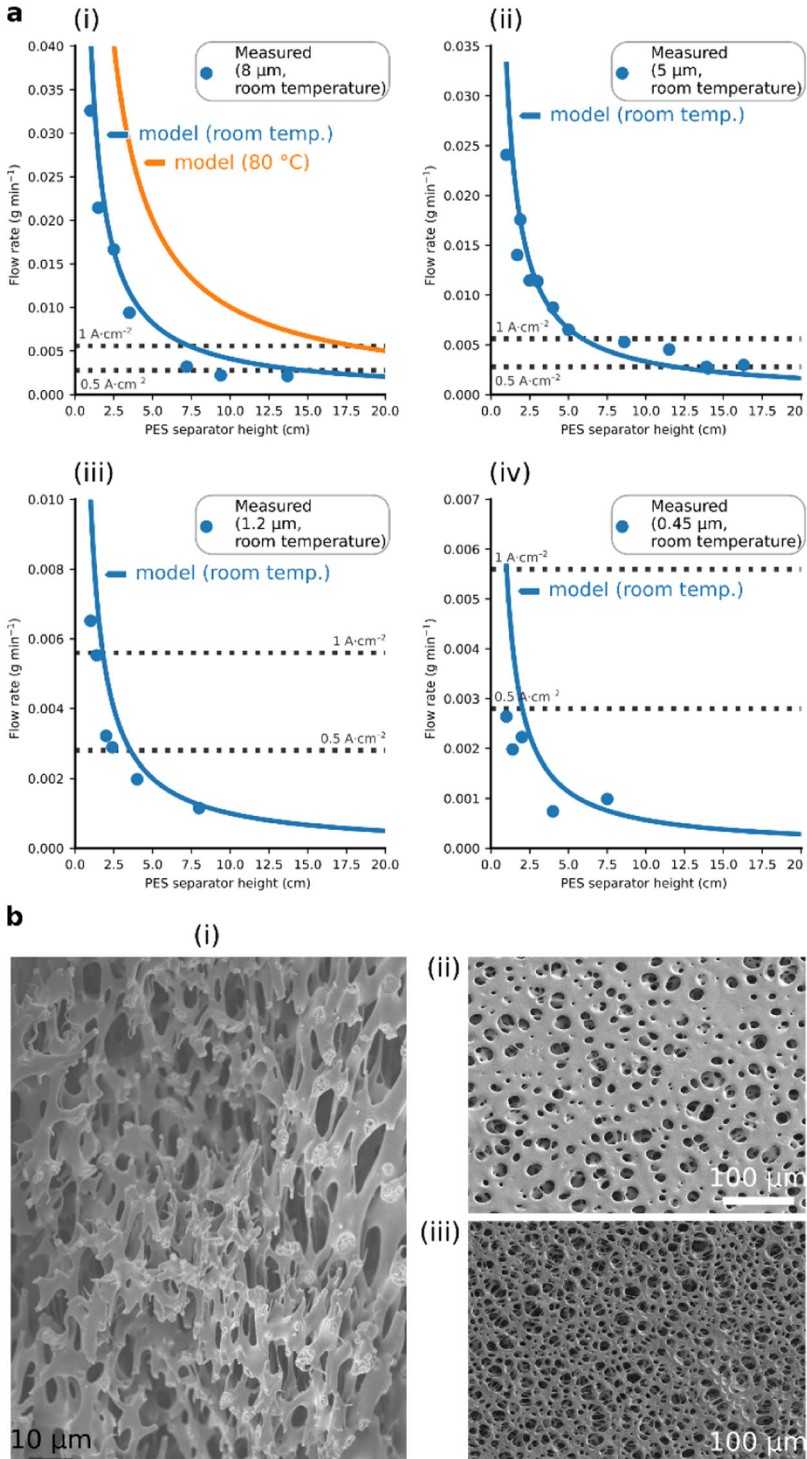

**Fig. 2 Capillary-fed inter-electrode separator. a** Flow rates at different heights inside a capillary-fed inter-electrode separator. Measured (data points) and modelled (blue line) flow rates (Darcy flow), at room temperature, of 27 wt% aqueous KOH within polyether sulfone (PES) separators that were specified as having average pore diameters of: (i) 8 μm, (ii) 5 μm, (iii) 1.2 μm, and (iv) 0.45 μm. The measured data was obtained as described in the Methods section and Supplementary Fig. 4. The modelled data was obtained as described in the text. The orange line in (i) depicts the modelled flow at 80 °C. The dotted lines show the rate of water consumption by a 1 cm$^2$ water electrolysis cell operating at 0.5 A cm$^{-2}$ (lower dotted line) and 1 A cm$^{-2}$ (upper dotted line). The capillary-induced flow rate within the 8 μm polyether sulfone separator is sufficient to supply water electrolysis at 0.5 A cm$^{-2}$ at a height of 15 cm at room temperature, and at 1 A cm$^{-2}$ at a height of 18 cm at 80 °C. **b** Pore structure of the polyether sulfone separator. Scanning electron micrographs of the polyether sulfone filter, showing: (i) its structure in cross-section, and (ii) its gloss surface and (iii) its matte surface.

| Table 1 Ionic resistance of separators at room temperature. | | | | |
|---|---|---|---|---|
| **Separator** | **Electrolyte** | **Thickness (μm)** | **Porosity (%)** | **Ionic Resistance (mΩ cm²)** |
| PES (8 μm) | 27 wt% KOH | 140 | 80 | 46 |
| Zirfon PERL UTP[18] | 30 wt% KOH | 500 | 55 | 290 |
| Nafion 115[19] | Hydrated | 125 | 0 | 142 |
| Nafion 117[19] | Hydrated | 183 | 0 | 220 |

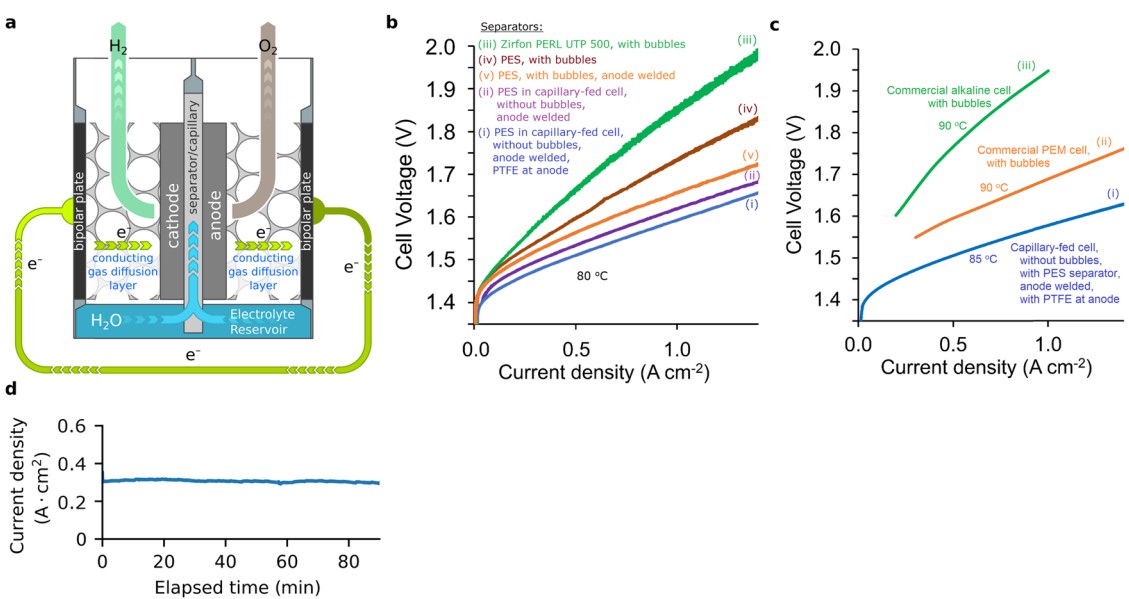

**Fig. 3 Capillary-fed electrolysis cell. a** Schematic depiction showing how the bipolar plate and conducting gas diffusion layer in the capillary-fed cell were combined into a single bipolar plate structure that comprised a sheet of Ni with many small holes to allow evolved gases to exit the electrode. The anode electrode was welded to its bipolar plate. The cathode was compressed against its bipolar plate and not welded. Supplementary Fig. 6 provides a picture of the cell used. **b** Polarisation curves of capillary-fed electrolysis cell and controls (80 °C). Plots of (2-electrode) cell voltage vs. current density, from bipolar plate to bipolar plate across the cell, excluding cathodic oxygen depolarisation, of cells with identical NiFeOOH anodes and Pt/C cathodes, at 80 °C, of: (i) Capillary-fed cell with PTFE at the anode, (ii) Capillary-fed cell without PTFE at the anode, (iii) control, conventional, bubbled cell with Zirfon PERL UTP 500 as inter-electrode separator, (iv) control, conventional, bubbled cell with polyether sulfone filter (8 μm average pore diameter) as inter-electrode separator, and (v) control, conventional, bubbled cell with polyether sulfone filter (8 μm average pore diameter) as inter-electrode separator, where the anode is welded to its bipolar plate. **c** Comparison of capillary-fed electrolysis cell (85 °C) with commercial alkaline and PEM electrolysis cells (90 °C) representative of the state-of -the-art. Polarisation curves of: (i) the capillary-fed electrolysis cell with PTFE at the anode at 85 °C, (ii) commercial alkaline electrolysis cell at 90 °C [24] representative of the state-of-the-art, and (iii) commercial PEM electrolysis cell at 90 °C[20] representative of the state-of-the-art. **d** Performance of capillary-fed electrolysis cell at a fixed 100% energy efficiency (HHV) (85 °C). (i) Current density at 85 °C of the capillary-fed electrolysis cell in Fig. 3c(i) when poised at a fixed cell voltage of 1.47 V, which equates to 100% energy efficiency (HHV). Supplementary Fig. 10 provides data from 1-day and 30-day tests at 80 °C and room temperature, with a large reservoir that was regularly manually replenished.

Figure 3a depicts a schematic of the cell. Polyether sulfone filters of much smaller pore diameter have been previously tested as inter-electrode separators in water electrolysis cells[21,22].

In commercial systems, bipolar plates are normally employed to carry current into and out of electrolysis cells. Within a cell, each bipolar plate connects electrically to their corresponding electrode via a conducting gas diffusion layer (also called a porous transport layer in PEM electrolysis cells) (Fig. 3a). The present work sought to replicate this arrangement to make fair comparisons with commercial cells. This was done by combining the bipolar plate and conducting gas diffusion layer into a thick sheet of Ni with many small holes to allow evolved gases to exit (Supplementary Fig. 6). This bipolar plate/flow field structure was housed within a gas chamber that could be sealed to the external environment. Leak-tight bolts through the walls of the gas chambers were used to press the bipolar plates and their attached electrodes against the polyether sulfone separator (Supplementary Fig. 6). The gas chambers were flushed with nitrogen prior to operation.

For the anode, a fine Ni mesh was electrocoated with a NiFeOOH electrocatalyst, as previously described by Benedetti and colleagues[23] and then spot-welded to its bipolar plate, without loss of catalyst, using the approach depicted in Supplementary Fig. 7. No carbon was present in the anode to avoid oxidative carbon corrosion currents, which can lead to highly misleading results.

In some cells, a 60 wt% dispersion of polytetrafluoro-ethylene (PTFE, also known as Teflon™) was included in the electrocoating solution. Catalyst coatings from the resulting solution were found to display enhanced anode performance.

For the cathode, a Pt/C electrocatalyst was deposited as previously described by Liu et al.[24], to a loading of 0.5 mg cm$^{-2}$ Pt on a conducting, carbon paper gas diffusion layer. As the carbon paper could not be welded, the cathode was pressed tightly against its bipolar plate as described in Supplementary Fig. 6. The structure of the cathode is described in detail in Supplementary Note 3.

**Table 2 Performance at 85 °C of the capillary-fed electrolysis cell with PTFE at the anode.**

| Current density (A cm$^{-2}$) | Cell voltage (V) | Energy efficiency (HHV) (%) | Energy consumption | |
|---|---|---|---|---|
| | | | (kWh kg$^{-1}$ H$_2$) | (kWh Nm$^{-3}$ H$_2$) |
| 0.294 | 1.470 | 100 | 39.4 | 3.55 |
| 0.500 | 1.506 | 98 | 40.4 | 3.64 |
| 0.800 | 1.551 | 95 | 41.6 | 3.75 |
| 1.000 | 1.575 | 93 | 42.2 | 3.80 |

Two-electrode measurements, including the polarisation curves, were recorded between the anodic and cathodic bipolar plates, across the cell, as they would be in a commercial cell. The electrolyte was 27 wt% aqueous KOH.

The current densities reported here are relative to the geometric area of the electrodes that were covered with electrocatalysts. Prior to testing, the current produced by the metal structures in the cell that were not coated with catalyst, was determined by operating the cell without catalysts, with and without the gas chambers filled with air. Currents of <0.025 A cm$^{-2}$ were observed up to a cell voltage of 2.1 V.

With catalysts, and with the gas chambers initially filled with air, oxygen depolarisation of the cathode occurred only up to 0.030 A cm$^{-2}$, whereafter hydrogen production overwhelmed oxygen ingress (Supplementary Fig. 8).

The CFE cell with the above electrodes and polyether sulfone separator was then tested at 80 °C, giving the current-voltage curves in Fig. 3b(i) (with PTFE in the anode) and Fig. 3b(ii) (without PTFE in the anode) (Source Data—Figs. 2–4). As can be seen in Fig. 3b(i), with PTFE incorporated in the anode, the cell required a voltage of 1.59 V to drive water electrolysis at 1 A cm$^{-2}$. Without PTFE in the anode, 1.61 V was needed at 1 A cm$^{-2}$ (Fig. 3b(ii)).

For comparative purposes, and to investigate changes in the cell resistance, identical zero-gap water electrolysis cells that were fully flooded with liquid electrolyte, causing them to produce gases in the form of bubbles, were also prepared. These control cells employed the same cathode and anode (without PTFE) as above, with 27 wt% KOH. They were tested at the same temperature of 80 °C.

With the well-known commercial alkaline separator, Zirfon PERL UTP 500, the resulting, bubbled control cell required 1.86 V to produce 1 A cm$^{-2}$ (Fig. 3b(iii)).

When the Zirfon was replaced with the polyether sulfone separator, the cell needed 1.74 V at 1 A cm$^{-2}$ (Fig. 3b(iv)), which was 0.12 V lower. This equates to a decrease in the cell resistance of 120 mΩ cm$^2$, which can be almost entirely attributed to the lower ionic resistance of the polyether sulfone separator compared to Zirfon PERL UTP 500 at 80 °C (Supplementary Fig. 5).

Following operation, a contact resistance was found to have developed between the anode and its bipolar plate. This contact resistance was due to the formation of a layer of poorly conducting Ni oxide on the contacting Ni surfaces by the oxygen produced at the anode.

To overcome this contact resistance, the control cell with the polyether sulfone separator, was modified by welding its anode to its corresponding bipolar plate. The anodes of the capillary-fed cells had been welded to their bipolar plates for the same reason. The effect was to decrease the voltage required at 1 A cm$^{-2}$ to 1.66 V (Fig. 3b(v)), which equates to a further decline of 0.08 V and an 80 mΩ cm$^2$ lower cell resistance. This result is relevant insofar as some commercial water electrolysis cells still employ

compression to create electrical contact[2], rather than welding, which is routinely used in other types of electrolysis cells[25].

The contact resistance between the cathode and its bipolar plate after operation was separately measured to be a much lower 3–5 mΩ cm$^2$. The reducing environment of the cathode avoids formation of poorly conducting surface layers.

Accordingly, the capillary-fed cell in Fig. 3b(i) significantly outperformed its conventional, bubbled, control cells employing the same electrodes and electrolyte in an identical zero-gap configuration.

The performance of the above CFE cell with PTFE at the anode, was also compared with data from commercial alkaline and PEM electrolysis cells representative of the current state-of-the-art[20,26]. As the commercial data had been collected at a higher temperature of 90 °C, the CFE cell was tested at 85 °C.

As can be seen in Fig. 3c, the CFE cell substantially outperformed the commercial cells. It improved on alkaline cells, which are typically operated commercially[2] at ~0.2–0.8 A cm$^{-2}$. It also improved on commercial PEM cells, which must be operated at higher current densities in the 1.5–3.0 A cm$^{-2}$ range to be commercially viable[20].

To produce typical commercial operating current densities of 0.5 A cm$^{-2}$ for alkaline and 1.8 A cm$^{-2}$ for PEM, the commercial cells required ~1.77 V (Fig. 3c(ii)–(iii)), equating to a cell energy efficiency of ~83% (HHV) and an energy consumption of ~47.5 kWh kg$^{-1}$ H$_2$. By contrast, the alkaline CFE cell required only 1.506 V at 0.5 A cm$^{-2}$ (Fig. 3c(i)), which represents a cell energy efficiency of 98% (HHV) with consumption of only 40.4 kWh kg$^{-1}$ H$_2$, or 3.64 kWh Nm$^{-3}$ H$_2$ (Table 2). The ~7.1 kWh kg$^{-1}$ decrease in energy consumption surpasses the IRENA 2050 target[2] of <42 kWh kg$^{-1}$ and realises a 15% improvement in cell energy efficiency.

When held at the thermoneutral voltage for water electrolysis, 1.47 V at 85 °C, which equates to 100% energy efficiency (HHV), the capillary-fed cell produced a constant ~0.3 A cm$^{-2}$ (Fig. 3d). As far as the authors are aware, no water electrolysis cell, either alkaline or PEM, has ever produced 0.3 A cm$^{-2}$ at 100% energy efficiency (HHV).

The CFE cell also demonstrated sustained stable performance over extended periods from 1 working day to 30 days continuously at 80 °C and room temperature, respectively, with periodic replenishment of the consumed water to the reservoir (Supplementary Fig. 10). Water spontaneously migrated from the reservoir up the separator, possibly under an osmotic as well as a capillary impulse to counteract increases in the KOH concentration in the separator due to water consumption at the electrodes. No KOH build up or crystallisation was observed in or on the separator.

**The origin of the high performance of the capillary-fed electrolysis cell.** The unprecedented performance of the CFE cell in Fig. 3b(i) and Fig. 3c(i) may be explained in part by the well-known high activity of the NiFeOOH anode electrocatalyst[25,27–30] and Pt/C cathode electrocatalyst, and the low ionic resistance of the polyether sulfone separator (Table 1).

Galvanostatic electrochemical impedance spectroscopy could be used to determine the resistances in the capillary-fed cell under active water electrolysis, but not their origin. For example, at 0.35 A cm$^{-2}$ and 80 °C, the cell demonstrated a series resistance of ~40 mΩ cm$^2$ that was mainly but not completely attributable to the low ionic resistance of the polyether sulfone separator (Supplementary Fig. 11).

To elucidate the major elements that contributed to the high performance of the CFE cell at 80 °C, the control, conventional, bubbled cell with Zirfon PERL UTP 500 separator (Fig. 3b(iii))

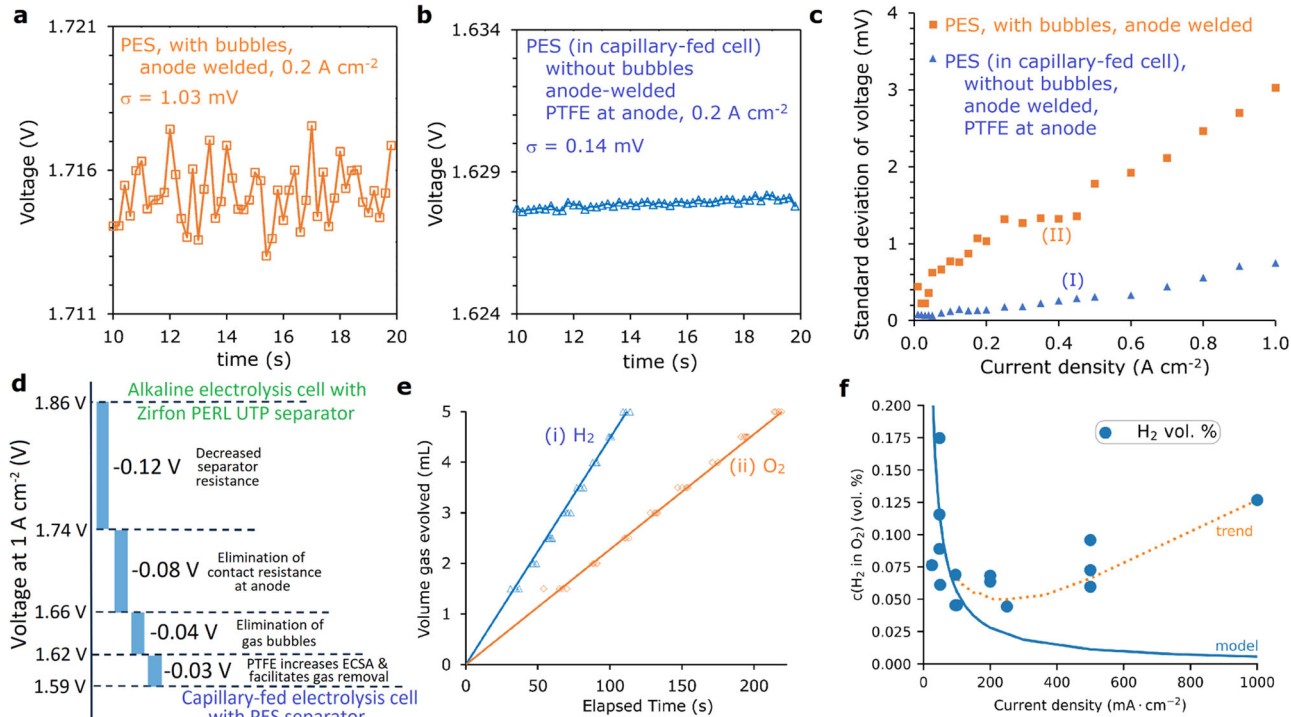

**Fig. 4 Capillary-fed electrolysis cell characterisation.** (At room temperature with 27 wt% KOH unless stated otherwise). **a** Voltage profile due to bubble formation. Voltage fluctuations in the bubbled, control cell in Fig. 3b(v) over the last 10 s of a 20 s step at 0.2 A cm$^{-2}$ ($\sigma$ = standard deviation). **b** Voltage profile in the capillary-fed cell. Voltage fluctuations in the cell in Fig. 3b(i) over the last 10 s of a 20 s step at a fixed 0.2 A cm$^{-2}$. **c** Voltage profile uniformity as a function of current density. Plot of the standard deviation in voltage for the cells in: (I) Fig. 3b(i), and (II) Fig. 3b(v), at different current densities, each held for 20 s, with the standard deviation measured over the last 10 s of the 20 s period. The bubbled control cell in Fig. 3b(v) displays much larger fluctuations in voltage due to bubble formation than the capillary-fed cell in Fig. 3b(i), which exhibits largely bubble-free operation at ≤0.2 A cm$^{-2}$, and substantially bubble-free operation at 0.25–1 A cm$^{-2}$. **d** Voltage declines and their origins. Waterfall plot showing the voltages declines observed at 1 A cm$^{-2}$ (80 °C) and their sources. **e** Faradaic efficiency. Rates of: (i) hydrogen generation, and (ii) oxygen generation by the capillary-fed electrolysis cell in Fig. 3b(i), at a fixed 0.350 A/cm$^2$ at atmospheric pressure, after 30 min. The data points indicate the measured volumes. The solid lines plot the theoretical rate of gas generation at 100% Faradaic efficiency. The overall Faradaic efficiency of the cell, determined by comparing the slopes of the measured and theoretical data, including both gases, was 99.5 ± 1.3%. **f** Hydrogen crossover. The data points show the hydrogen crossover into the anodic oxygen stream of the capillary-fed electrolysis cell in Fig. 3b(i) as a function of current density, at room temperature and atmospheric pressure. Each data point was collected after operating the cell for 30 min at the relevant current density. The solid line shows the gas crossover expected from diffusion only[31, 32]. The dashed line depicts the trend in the data.

was therefore taken as a baseline. That cell needed 1.86 V to generate 1 A cm$^{-2}$ (Fig. 3b(iii)).

When the Zirfon was replaced with the polyether sulfone separator, the voltage required at 1 A cm$^{-2}$ was 1.74 V (Fig. 3b(iv)), which was 0.12 V less. As noted, the resulting 120 mΩ cm$^2$ reduction in resistance is almost all due to the lower ionic resistance of the polyether sulfone separator.

The effect of welding the anode of the bubbled cell with the polyether sulfone separator, to its bipolar plate was to further decrease the voltage needed at 1 A cm$^{-2}$ to 1.66 V (Fig. 3b(v)). The resulting 80 mΩ cm$^2$ decrease in resistance arose by elimination of a contact resistance that developed between the anode and its bipolar plate during operation.

The CFE cell achieved still lower voltages. Without PTFE at the anode, it required a cell voltage of 1.62 V at 1 A cm$^{-2}$ (Fig. 3b(ii)), which is 0.04 V lower, equating to a further 40 mΩ cm$^2$ decrease in resistance.

With PTFE at the anode, the capillary-fed cell required only 1.59 V at 1 A cm$^{-2}$ (Fig. 3b(i)), which constituted a still further decrease of 0.03 V, with an additional lowering in cell resistance of 30 mΩ cm$^2$.

These improvements may be attributed to the combined contribution of several possible factors, including the following.

Firstly, the CFE cells were largely bubble-free during operation. This was confirmed by comparing voltage fluctuations due to

bubble formation and release at a series of fixed current densities[12]. Fig. 4a and b show the results and analysis of steady-state chrono-potentiometric measurements of the bubbled control cell in Fig. 3b(v) and the capillary-fed cell in Fig. 3b(i), respectively, over the last 10 s of a 20 s period at 0.2 A cm$^{-2}$. The voltage of the bubbled cell (Fig. 4a) (Source Data—Figs. 2–4) was characterised by a noisy response attributable to the nucleation, growth, coalescence, and release of gas bubbles. The standard deviation ($\sigma$) of the voltage signal for the bubbled cell was 1.03 mV. By contrast, the voltage of the capillary-fed cell under the same conditions was remarkably steady (Fig. 4b), with a significantly lower $\sigma$ of 0.14 mV. The steady voltage signal and low $\sigma$ value of the capillary-fed cell is consistent with bubble-free operation.

Steady-state chrono-potentiometric measurements of bubbled and capillary-fed cells over a range of current densities from 0.01 to 1 A cm$^{-2}$ and analysis of the results permitted the construction of the standard deviation vs. current density plot in Fig. 4c. The bubbled cell demonstrated a relatively steep increase in $\sigma$ values commencing from the lowest current density of 0.01 A cm$^{-2}$. By contrast, the capillary-fed cell had much lower and flatter $\sigma$ values between 0.01 and 0.2 A cm$^{-2}$, before the $\sigma$ values increased modestly as the current density was raised to 1 A cm$^{-2}$. The results suggest that the capillary-fed cell was largely bubble-free

up to and including 0.2 A cm$^{-2}$, and substantially bubble-free between 0.25 and 1 A cm$^{-2}$. At 1 A cm$^{-2}$, the capillary-fed cell displayed a $\sigma$ value of 0.75 mV, which was comparable to the bubbled cell at ~0.09 A cm$^{-2}$, suggesting that <10% of the current went into gas production via bubble formation (Supplementary Fig. 9).

These findings are supported by the fact that the performance of the capillary-fed cell in Fig. 3c(i) mainly differed from the commercial PEM cell in Fig. 3c(iii) in having a lower onset potential; the slopes of the curves were similar. Previous studies have demonstrated that bubble-free operation decreases the onset potential[10,11,13]. The bubble-producing cells in Fig. 3b(iii)–(v) had onset potentials of ≥1.45 V, while that of the capillary-fed cell in Fig. 3b(i) was ~1.39 V.

Bubble-free gas evolution likely contributed to the voltage decline insofar as the electrodes were not masked with bubbles, leaving the catalytic sites on the electrodes more available for reaction. This probably enhanced the overall performance of, particularly, the most active catalytic sites that are the source of most bubbles and the first to be blocked by bubbles. The resulting fuller use of the available electrocatalytic sites likely improved performance.

The avoidance of bubble formation at ≤0.2 A cm$^{-2}$, which was likely due to the gas-liquid interface being within diffusion distance of the electrode, may also have decreased the super-saturation of the electrolyte, leading to a voltage decline[10,13]. According to the Nernst equation, elevated gas concentrations increase E°[33]. At higher current densities, supersaturation may have been needed at some electrode locations to produce the few bubbles observed.

The architecture of the capillary-fed cell may also have contributed insofar as it ensured that the flow of water toward each electrode did not counter the flow of gas away from the electrode. That is, the architecture of the capillary-fed cell inherently avoided the counter multiphase flows present in conventional, bubbled water electrolysers.

At this stage it is not possible to determine the absolute contribution of each of the above factors, but cumulatively they resulted in a 40 mΩ cm$^2$ lower resistance in the capillary-fed cell without PTFE at the anode (Fig. 4d).

With PTFE at the anode, additional improvements were realised in the capillary-fed cell (Fig. 3b(i)). Electron micrographs indicated that the PTFE was dispersed as needle-like structures on the anode (Supplementary Fig. 12). The double layer capacitance of the anode increased by ~12-fold, from 0.46 mF cm$^{-2}$ without PTFE to 5.50 mF cm$^{-2}$ with PTFE (Supplementary Fig. 13). This suggests that the electrochemically/catalytically active surface area (ECSA) of the anode increased significantly. As the specific capacitance of the catalyst with PTFE is not known, the precise scale of the increase could not be determined. However, the PTFE clearly increased the porosity of the electrocatalytic layer and this increased the ECSA.

A body of previous work has also demonstrated that PTFE surfaces on an electrocatalyst may scavenge, coalesce and transport away newly formed, dissolved gases[34]. PTFE is highly aerophilic, with low surface energy. A similar mechanism may have been partly responsible for the improved, bubble-free performance of the anode when PTFE was incorporated. That is, the PTFE on the anode may have amplified the catalytic performance by facilitating migration of newly formed gas along its aerophilic surfaces, across the gas-liquid interface, to thereby avoid gas bubble formation. As described in Supplementary Note 3, it is potentially significant that the same elements of aerophilic PTFE surface pathways for gas transport across the gas-liquid interface were also present on the cathode, which exhibited similarly bubble-free performance.

Whatever the origin of the improved performance, the incorporation of PTFE in the anode contributed a decrease in voltage of 0.03 V at 1 A cm$^{-2}$, equating to a decline in resistance of 30 mΩ cm$^2$ (Fig. 4d).

In summary, the capillary-fed cell continues the evolution of water electrolysis cells by systematic decreases in cell resistance, as illustrated in Fig. 1. However, the large net decrease realised did not have a single origin. It involved many smaller contributions that, cumulatively, led to a 270 mΩ cm$^2$ reduction in cell resistance at 80 °C, over the standard commercial configuration of a bubbled, zero-gap cell with a Zirfon PERL UTP 500 separator (Fig. 4d).

**Faradaic efficiency approaching 100%, and low hydrogen crossover**. The Faradaic efficiency and gas crossover are important features of electrolysis cells. The latter is a potential safety issue as a hydrogen stream containing >4.6% oxygen, or an oxygen stream with >3.8% hydrogen, constitutes an explosive mixture (at 80 °C)[10].

The volumes of hydrogen and oxygen produced by the best performing capillary-fed cell at a fixed 0.35 A cm$^{-2}$ were measured (Fig. 4e). There was a close agreement between the measured volumes and what would be expected if all electrons went into water electrolysis, giving an overall Faradaic efficiency, including both gases, of 99.5 ± 1.3% (Source Data—Figs. 2–4).

To measure the extent of hydrogen gas crossover, the CFE cell with PTFE at the anode was connected to a gas chromatograph and measurements were taken of the hydrogen impurity in the anodic oxygen stream. Figure 4f shows the concentration of hydrogen in the anodic oxygen stream as a function of current density. These results, which fell between 0.04 and 0.14 vol% at 0.1–1.0 A cm$^{-2}$, are notably lower than reported rates of hydrogen crossover with conventional separators (Supplementary Table 5).

The oxygen impurity in the cathodic hydrogen stream was similarly examined, however no crossover could be observed within the detection limit of 0.001 vol% (10 ppm).

The low crossover of the capillary-fed cell may be ascribed to a different and unique mechanism of gas crossover.

In fully flooded, bubbled alkaline electrolysis cells, gas cross-over is known to occur in two ways[34,35]: (1) diffusion of dissolved gas through the liquid in the separator, and (2) advective flow of liquid electrolyte, carrying dissolved gas and bubbles with it, through the porous separators that are generally used. The latter is driven by fluctuating or perpetual pressure differentials across the separator, and is, by far, the larger contributor, producing orders of magnitude more crossover[35,36]. To decrease the advective cross flow of electrolyte, separators in alkaline electrolysis are designed to have the smallest possible pores (<0.15 μm diameter)[35]. The rate of diffusion-based crossover in alkaline cells is exceedingly low because the high levels of K$^+$ and OH$^-$ ions in alkaline electrolytes salt-out dissolved gases like hydrogen and oxygen, which have very low solubilities and diffusion coefficients in alkaline electrolytes[31,32].

In PEM electrolysis cells, the proton exchange membranes are non-porous. This eliminates advective flows as a significant mechanism of gas crossover since the de-ionised water used in such cells is unable to freely pass through the membrane[36,37]. The only available mechanism of gas crossover is diffusion[3]. The combined solubility and diffusion coefficients of hydrogen and oxygen are, however, 40–120-times higher in de-ionised water than in typical alkaline electrolytes at 80 °C[31,32,36]. While advective crossover is absent in PEM cells, diffusion-based crossover, with balanced hydrogen and oxygen pressures on opposite sides of the PEM membrane, is therefore usually notably

larger than in alkaline systems (Supplementary Table 5)[36–38]. The industry has overcome this issue by using very high hydrogen pressures but only atmospheric oxygen pressures on the opposite sides of the PEM membrane. This maintains the produced hydrogen free of oxygen impurity[37].

The CFE cells are in the unique position of avoiding advective crossover, whilst also having low diffusion-based crossover because of the high molarity alkaline electrolyte used.

Advective crossover is not available as a mechanism of crossover in capillary-fed cells since there are no unrestricted bodies of liquid electrolyte on both sides of the separator that are free to flow through it under the influence of a pressure differential. That is, only diffusion-based crossover is possible. However, a high molarity alkaline electrolyte is used, and this provides for only a small diffusion-based crossover.

This explanation is supported by the fact that, whereas separators in conventional, fully flooded, bubbled alkaline electrolysis cells can only minimise crossover by having very small pores (<0.15 μm), the capillary-fed cell has the largest of the available pores (8 μm) but still exhibits minor gas crossover.

It is further supported by the observation in Fig. 4b, that, at current densities of ≤0.2 A cm$^{-2}$, the measured rates of gas crossover lie within the range expected for diffusion only, which is depicted as the solid line. At current densities above 0.2 A cm$^{-2}$ however, hydrogen crossover diverges from the diffusion-only model and trends moderately upward (Fig. 4f, orange dotted line). This is consistent with the observed minor bubble formation above 0.2 A cm$^{-2}$ in Fig. 4c, which may involve supersaturation at a few locations on the hydrogen-generating cathode[36]. Such localised supersaturation would create a partial pressure gradient resulting in moderately increased hydrogen crossover.

**Simplification of the balance-of-plant**. Capillary-fed electrolysis (CFE) cells may be readily incorporated into a bipolar cell stack of the type used in commercial electrolysers (Supplementary Fig. 2). The engineering system required to manage such a stack, known as the balance-of-plant, may then be compared with a typical, conventional balance-of-plant[13].

Supplementary Fig. 3 depicts this comparison. As can be seen, the balance-of-plant needed for the CFE cell concept (Supplementary Fig. 3b) is notably less complex than that needed for a conventional, bubbled electrolysis cell (Supplementary Fig. 3a).

There is, firstly, no need for pumped liquid circulation, or for gas/liquid separators, as there are no liquid-enveloped gas bubbles that need to be constantly swept away from the electrodes (Supplementary Fig. 3b). The gas-liquid froths that are generated when gas bubbles are produced must normally be pumped to separator tanks for partitioning into bulk liquid and gas phases (Supplementary Fig. 3a).

The high-volume electrolyte pumps (depicted below each separator tank in Supplementary Fig. 3a) are therefore also not needed, nor are their associated piping and fittings, namely, the anolyte and catholyte forward and return lines. Pumps and piping of this type are expensive because they need to avoid corrosion by KOH (in the case of an alkaline electrolyser) or leaching of metal ions into the de-ionised water used (in the case of a PEM electrolyser) and comply with stringent hydrogen or oxygen safety standards.

In a conventional balance-of-plant, de-ionised make-up water would typically be added to each electrolyte circulation loop, via the scrubbers, from a pressurised dispensing system (as shown at the top of Supplementary Fig. 3a). In the capillary-fed balance-of-plant, a comparable pressurised dispensing system would be needed to add de-ionised water to the individual reservoirs (as shown at the bottom of Supplementary Fig. 3b). Such arrangements are already used in the chlor-alkali industry, where make-up water and brine are routinely dispensed to individual half cells[39].

Another feature of conventional balance-of-plants are the need for water-cooled chillers to remove the excess heat produced during electrolysis (Supplementary Fig. 3a). The high energy efficiency of the capillary-fed cell produces only modest Joule heating during operation however, avoiding the need for a water-cooled chiller as demonstrated in Supplementary Note 1, Supplementary Tables 1–2, and Supplementary Data 1. Instead, air-cooling or radiative self-cooling of the stack may be possible.

Because of the need for large volumes of liquid to remove and separate the gas bubbles formed, conventional commercial cell stacks and their balance-of-plants typically contain ~10,000 L of water per MW. By contrast, as shown in Supplementary Note 2 and Supplementary Table 3, a, the CFE system is likely to require only ~500 L of water per MW.

One effect of a lower water requirement is a reduced need for water purification and replacement in the balance-of-plant. This is especially relevant to PEM electrolysers, which require on-going de-ionisation of the water content, using a costly class 1 de-ioniser, to achieve a stack lifetime of ~70,000 h. In alkaline electrolysers, only the input make-up water needs deionisation, with the entire body of liquid electrolyte typically needing replacement every 5–6 years[8].

A final advantage of the capillary-fed cell system is its capacity to avoid the wasteful and corrosive high voltage shunt currents[13] that flow between cells along the catholyte and anolyte return lines in conventional alkaline electrolysers. In the capillary-fed balance-of-plant, the only potential liquid conduction path between cells is along the water dispensing line shown at the bottom of Supplementary Fig. 3b. But this contains de-ionised water that is not conductive.

These simplifications in the balance-of-plant lead to downward pressure on electrolyser CAPEX. They may also lower the power consumption of the balance-of-plant, further decreasing the energy needed per kg hydrogen.

## Discussion

This work introduced the CFE cell concept. Using existing catalysts, with Faradaic efficiencies approaching 100%, and low hydrogen crossover, this architecture significantly improved the energy efficiency of the water electrolysis cell. At the operating current density used in many commercial alkaline electrolysis cells of 0.5 A cm$^{-2}$, a cell voltage of only 1.506 V was needed to produce hydrogen at 85 °C. This represents a cell energy efficiency of 98% (HHV), with consumption of just 40.4 kWh kg$^{-1}$ H$_2$, or 3.64 kWh Nm$^{-3}$ H$_2$. This result surpasses commercial electrolysis cells, which consume ~47.5 kWh kg$^{-1}$ H$_2$, and exceeds the 2050 IRENA target[2] of <42 kWh kg$^{-1}$.

The CFE cell also allows for a notably simplified balance-of-plant, further reducing energy consumption and putting downward pressure on CAPEX.

These substantial improvements on present-day state-of-the-art electrolysis cells translate to direct declines, of similar proportion, in the levelised cost of hydrogen. Combined with the promise of a simplified system balance-of-plant, they bring cost-competitive renewable hydrogen closer to reality.

## Methods

**Materials**. Porous polyether sulfone (PES) filters of 0.03 μm, 0.45 μm, 1.2 μm, 5 μm, and 8 μm average pore diameters (Pall Corporation and Sterlitech), Carbon black (AkzoNobel), 10% Pt on Vulcan XC-72 (Premetek), NiCl$_2$.6H$_2$O and FeCl$_2$.4H$_2$O (Sigma-Aldrich), PTFE (60 wt.% dispersion in alcohols/H$_2$O; Sigma-Aldrich, 510211), Nafion™ dispersion (5% in in alcohols/water; Sigma-Aldrich),

Sigracet™ 22BB carbon paper (Fuel Cell Store), Zirfon PERL UTP 500 (Agfa), KOH 90%, flakes (Sigma-Aldrich), Ni mesh (Century Woven, Beijing).

**Porosity and in-plane flow model values**. Porosity was determined by comparing the weight of dry and saturated polyether sulfone. Cross-sectional area was calculated by multiplying width (1 cm, nominal) by thickness (140 μm, measured). Average pore diameter was determined using capillary flow porometry. Contact angle was measured using the captive bubble technique. Viscosity[16] and surface tension[17] were obtained from the literature.

**In-plane flow rate measurements**. In-plane flow rates were measured using the setup shown in Supplementary Fig. 4(a). For each experiment a 1 cm-wide polyether sulfone microfiltration strip was encased in plastic sheathing of length $L$ to avoid electrolyte evaporation; 1.5 mm at each end of the polyether sulfone strip was left exposed. To the top of the strip was clamped a pad composed of several layers of absorbent paper. The bottom of the strip was dipped in electrolyte. The weight of the assembly was measured over time. An example weight vs. time plot is provided in Supplementary Fig. 4(b) and shows an initial curved response corresponding to initial filling of the strip (Washburn flow), followed by a linear response corresponding to continuous flow through the fully wetted strip (Darcy flow). The in-plane flow rate was taken as the slope of the linear part of the weight vs. time plot.

**Ionic resistance of polyether sulfone separator**. The ionic resistance of the polyether sulfone separator was determined using a method described in the literature[40] and the setup shown in Supplementary Fig. 5. The conductivity of the KOH electrolyte was measured with and without a polyether sulfone separator present using a four-point conductivity probe (Mettler Toledo Sevencompact with Inlab ISM-731 Probe) as described in Supplementary Fig. 5.

**Electrode preparation**. The hydrogen-evolving cathode was prepared using a literature method[24]. 100 mg 10% Pt on Vulcan XC-72, 0.8 mL 5 wt% Nafion solution, 1.5 mL deionised water, and 3 mL iso-propanol were combined and sheared at 10,000 rpm for 5 min using a homogeniser (IKA T25). The resulting dispersion was air brushed onto the microporous layer side of a 1 cm × 1 cm carbon fibre paper (Sigracet 22BB) to a loading of 0.5 mg cm$^{-2}$ Pt.

The oxygen-evolving anode was prepared following a literature method[23]. Prior to use, Ni mesh (200 LPI, ø 50 μm wire, 75 μm aperture) was cleaned by ultrasonication in isopropyl alcohol for 10 min and dried, then pickled in 5 M HCl for 10 min, rinsed with deionised water, and dried. After taping as shown in Supplementary Fig. 7, electrodeposition was performed at room temperature in a 3-electrode cell comprising 1 cm × 1 cm Ni mesh working electrode, oversized Ni mesh counter electrode, and Ag/AgCl (3 M NaCl) reference electrode. The nickel mesh was placed in an electrocoating solution that comprised of a 3:1 mixture of NiCl$_2$.6H$_2$O (0.075 M) and FeCl$_2$.4H$_2$O (0.025 M) (following Fig. 8(c) and Fig. 1(a) in ref. [23]), along with 1 M KCl supporting electrolyte (following Fig. 8(b) in ref. [23]). The nickel mesh immersed in the electrocoating solution was coated with NiFe by repeated cycling using cyclic voltammetry between −1.0 V and −0.2 V (vs. Ag/AgCl) at 10 mV s$^{-1}$ until a charge of 17 C had been deposited (following Fig. 1 in ref. [23]). A BioLogic VSP potentiostat was used. The lower voltage of −1.0 V allowed for inclusion of a PTFE dispersion, without precipitation; the upper voltage of −0.2 V was found to provide the best catalytic performance. A dispersion of PTFE could be incorporated into the anode by including the equivalent of 10 g L$^{-1}$ PTFE in the electrocoating solution. Following electrodeposition, the working electrode was rinsed with deionised water and dried at room temperature. The electrocoating tape was then removed and the anode was welded to the bipolar plate as shown in Supplementary Fig. 7.

**Capillary-fed cell**. A picture of the capillary-fed cell used experimentally is provided in Supplementary Fig. 6. The polyether sulfone separator (8 μm average pore diameter) was initially soaked in deionised water and then kept in 27 wt% KOH electrolyte overnight. Ni gas diffusion bipolar plates measured 1.4 cm × 1.4 cm × 0.1 cm and included numerous 1 mm holes to allow evolved gases to exit the electrode. They performed the function of both the bipolar plate and the gas diffusion layer/porous transport layer in electrolysis cells. Bare Ni mesh was spot-welded to the Ni gas diffusion-bipolar plate on the cathode side and then pressed tightly against the rear of the Sigracet cathode substrate. The contact resistance was 3–5 mΩ cm$^2$. The Ni mesh anode was spot-welded to the Ni gas diffusion bipolar plate on the anode after coating with NiFe catalyst as described in Supplementary Fig. 7. The capillary-fed cell had the architecture: Ni bipolar plate/Ni mesh/Pt-C on carbon paper/ polyether sulfone separator/NiFe-coated Ni mesh/Ni bipolar plate. The anode and cathode were pressed against the polyether sulfone separator by leak-tight bolts passing through the gas chambers (Supplementary Fig. 6). Finally, the reservoir into which the bottom end of the polyether sulfone separator was dipped, was filled with 27 wt% KOH electrolyte.

**Electrochemical measurements**. Electrochemical measurements were performed using a BioLogic VMP3 potentiostat. Linear sweep voltammetry was performed by sweeping the cell voltage upward from 1.2–1.4 V at 10 mV s$^{-1}$ until the current

density reached 1.5 A cm$^{-2}$. Measurements at 80 °C or 85 °C were performed upon temperature equilibration after placing the capillary-fed cell into an oven at that temperature. Galvanostatic electrochemical impedance spectroscopy (GEIS) was performed at 0.350 A cm$^{-2}$ DC bias, 0.050 A cm$^{-2}$ AC perturbation, and between 100 kHz and 100 mHz.

**Faradaic efficiency**. Faradaic efficiency was calculated by comparing the measured volumes of hydrogen and oxygen produced at the cathode and anode, of a capillary-fed cell, respectively, at a fixed current of 0.350 A cm$^{-2}$ at room temperature, with the expected volumes of hydrogen and oxygen if all electrons resulted in gas evolution (including the expected water vapour content of the gases and considering the gas crossover, which was measured at the same time). Gas volumes were measured by collecting the produced gas in a submerged, upturned burette. The cell was operated for 30 min at 0.350 A cm$^{-2}$ prior to taking each set of measurements.

**Hydrogen crossover**. The concentration of hydrogen in the anode oxygen stream of a capillary-fed cell operated at different current densities at room temperature was determined by piping the anode gas output for analysis to a gas chromatograph (Shimadzu GC8A), operating with thermal conductivity detection and flame ionisation detector and argon as a gas carrier. The hydrogen and oxygen peaks were identified by their retention times and integrated to determine the relative quantities of gas present.

## Data availability

Source data are provided with this paper.

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

## Acknowledgements

The authors gratefully acknowledge support from the Australian Renewable Energy Agency (ARENA), Grant number DM015 (entitled: Ammonia production from renewables) (G.F.S. & G.G.W.). This activity received funding from ARENA as part of ARENAs Research and Development Program-Renewable Hydrogen for Export. Support from the Australian Research Council Centre of Excellence Scheme (grant number CE140100012) (G.G.W. and others) and the Australian National Fabrication Facility (ANFF) Materials Node is also acknowledged. The authors acknowledge the assistance of the University of Wollongong Electron Microscopy Centre. This research used equipment funded by the Australian Research Council —Linkage, Infrastructure, Equipment, and Facilities grant LE160100063.

## Author contributions

A.H. and A.L.H. contributed equally to this work. A.H. developed the model for capillary-induced electrolyte flow in a porous inter-electrode separator, measured the ionic resistance of the separators and studied the Faradaic efficiency and gas crossover of capillary-fed electrolysis cells employing them. A.H. further developed the model for diffusion-based gas crossover referred to. A.L.H. developed and studied all aspects of the electrodes, as reported herein, including the catalysts, their bubble-free operation, and detailed optimisation. G.T., K.W., C.Y.L., and G.W. co-supervised or aided the above work. G.S. was the primary supervisor. The project was conceived of by G.S. and K.W. G.S. and G.T. wrote and edited the paper, and all authors commented on it.

## Competing interests

G.T., K.W., C.Y.L., and G.S. are employees of the University of Wollongong but are presently, or will be in future, performing paid work for Hysata Pty Ltd, a company that has licensed the capillary-fed electrolysis cell technology. A.H., A.L.H., K.W., C.Y.L., and G.F.S. are co-inventors on patent applications covering the technology that have been filed. The remaining authors declare no competing interests.
