## [Peer Review File · Nature Communications]

A high-performance capillary-fed electrolysis cell promises more cost-competitive renewable hydrogenREVIEWER COMMENTS

Reviewer #1 (Remarks to the Author):

Review of the article "A high-performance capillary-fed electrolysis (CFE) cell promises more cost-competitive renewable hydrogen" by Aaron Hodges, Anh Linh Hoang, George Tsekouras, Klaudia Wagner, Chong-Yong Lee, Gerhard F. Swiegers, and Gordon G. Wallace.

This article details a very interesting novel configuration for water electrolysis, where the water is fed through the diaphragm allowing almost full bubble-free operation. The idea is elegant and very well described in the paper. Its various benefits are clearly explained and argued. In my opinion there is a lot of potential in the proposed approach, with benefits including a higher efficiency, simpler balance of plant, no shunt currents, and reduced gas cross-over. Some of the few downsides seem to be an electrode height limitation and some concern regarding whether heat can be removed effectively.

I can wholeheartedly recommend publication, as this constitutes important progress in the field. What follows are mostly minor comments, I hope the authors will consider, and a few more major suggestions. I would particularly like to hear argumentation, should the authors decide not to follow the first suggestion.

Congratulations with your excellent results and great paper.

Kind regards,

Willem Haverkort (j.w.haverkort@gmail.com)

PS: I may have a rudimentary understanding of where the voltage improvement of the bubble-free designs comes from, but hope you don't mind that I first further investigate and quantify the effect before I write it down properly.

major comments:

- An energy efficiency that can exceed 100 % does not make much sense. I would therefore be very much in favor of using the lower heating value of hydrogen as a reference, instead.

See also "Lamy, C., & Millet, P. (2020). A critical review on the definitions used to calculate the energy efficiency coefficients of water electrolysis cells working under near ambient temperature conditions. *Journal of Power Sources*, 447, 227350."

Eq. (21) of this reference, using the reversible potential of 1.23 V as a reference is a fine approximation. Insisting on using an efficiency based on the thermoneutral potential, the difference with the reversible potential should be added to the denominator, as in Eq. (20).

- "The capillary-fed electrolysis cell also demonstrated sustained stable performance over extended periods from 1 working day to 30 days continuously at 80 oC and room temperature, with periodic, manual, replenishment of the consumed water (Supplementary Fig. 10)."

Please provide the used current density (for the days figure) and voltage (for the hours figure).

There does seem to be a significant increase in voltage over the course of a month that seems to be swept under the rug. I would recommend to zoom in for the graphs of Supplementary Fig. 10 and comment on what could be the cause of this increase.

- "Instead, air-cooling or no cooling (i.e. radiative self-cooling of the stack) may be possible". It is not explained how the authors envision operating a scaled-up version at elevated electrolyte temperature without putting the electrolyzer into an oven. I envision that large temperature gradients will arise, with localized heating and little convective cooling. If instead operation very close to the thermoneutral voltage is envisioned there should be good insulation present?

- details for Supplementary Section 1 seem to be almost completely missing. Therefore I would recommend adding details on the performed calculations and the assumptions involved (or remove the section).

(The same holds true also for the LCOH calculations in Supplementary Figure 1, but there the calculation is arguable more straightforward).

major minor comments.

- use 27 wt% rather than % for clarity at several instances

- the value of the PES measured contact angle does not seem to be reported?

- "The solid line shows the gas crossover expected from diffusion only." please provide at least the assumed effective diffusivity and concentration difference.

- "At current densities above 0.2 A cm⁻², hydrogen crossover trends very moderately upward" The word "very" seems to downplay the very significant deviation (an order of magnitude difference between calculated and measured) too much and may be nuanced.

- If I understand correctly the "separator resistances" in table 1 are calculated by subtracting the ohmic drop of the electrolyte, which makes some sense, but may be non-standard? This resistance will still depend on the electrolyte conductivity, so is not a property of the separator. My impression is that most papers report the separator resistance including that due to the electrolyte (R_E), which makes more sense to me. Also, it is not made explicit how R_E is obtained, so there can still be some doubt over the exact procedure followed.

- it did not become fully clear to me whether the anode electrode contained the same Sigracet carbon paper as on the cathode.

minor minor comments:

- "to commit to net-zero carbon emissions by 2050 to limit global warming to 1.5 °C above pre-industrial levels" this 'commitment' is rather unbinding, and also the "to" is not a logical implication, so this sentence may be reconsidered.

-Please provide a reference for the statements

"The OPEX is, by far, the larger component of LCOH"

"which recently culminated in asymmetric polymer electrolyte membrane (PEM) cells that directly produce one of the gases in a gas collection chamber..."

and

"with the entire liquid electrolyte typically replaced every 5-6 years."

- "Any improvements in net energy efficiency create a similarly large decrease in the levelised cost of the produced hydrogen" similarly should be proportionally?

- The 8 and 4 in the denominator of Eqs. (1) and (2) should preferably appear first

- just above Eq. (5) it reads "equ." (2). Is this the journal preferred way of referring to an equation?

- "Catalyst coatings from the resulting solution were found to display enhance anode performance"
enhance > enhanced

- "As the carbon paper could not be welded, the cathode was pressed tightly against its bipolar plate"
How was this pressing done?

- "The current densities reported here are relative to the geometric area of the electrocatalysts." The use of the vague term "geometrical area" seems to leave still some room for confusion, especially when used in conjunction with "the electrocatalyst", perhaps change to "the geometrical area of the electrode that is covered with electrocatalyst"?

- The resolution of the figures is somewhat poor, especially the text. Perhaps consider using either vector drawings, or a higher resolution.

- "the capillary-fed cell displayed a σ value of 0.75, which was comparable to the bubbled cell at $\sim 0.09 \text{ A cm}^{-2}$ " 0.75 should be 0.75 mV

- "The combined solubility and diffusion coefficients of hydrogen and oxygen are, however, 40-120-times higher in de-ionized water than in typical alkaline electrolytes at 80 °C³³⁻³⁵. While advective crossover is therefore absent in PEM cells" The word "therefore" seems a bit misplaced?

- "The capillary-fed electrolysis cell also allows for a notably simplified the balance-of-plant" remove "the"

- The outcome of Supplementary Section 2, 398 liter, seems to have a few too many significant digits for such a crude estimate.

- Eqs. (6) and (7) of Supplementary Fig. 5 are already in the paper and do not seem to fulfil a particular purpose here, so can best be removed?

- Some interpretation of Supplementary Fig. 11 seems desirable.

- Regarding Supplementary Fig. 12 one is left wondering why the PTFE increases the double layer capacitance. Any useful comments there?

Reviewer #2 (Remarks to the Author):

The manuscript "A high-performance capillary-fed electrolysis (CFE) cell promises more cost-competitive renewable hydrogen" introduces a novel concept of the water electrolysis systems, which demonstrates very high performance, even exceeding commercially available devices. The authors clearly describe a bipolar stack of CFE cells as well as their balance of plant. The results shown are well supported and should be of wide interest in the scientific community due to discovering of several important contributions that cumulatively led to a significant decrease in the cell resistance compared to the standard configuration of alkaline water electrolysis cells. While reported performance appears outstanding, I have several questions or comments which should be clarified to further improve the quality of the research.

1) The authors utilize 27% KOH electrolyte but never clarified their choice. It is expected that the molarity of the electrolyte will affect the in-plane capillary-induced transport of a liquid through a porous material. Besides, since the water will constantly be consumed during electrolysis, changing of the KOH molarity in a thin layer close to the electrode surface might be expected. Could authors comment on this? How they managed to avoid crystallization of KOH at the upper side of the separator (or inside its porous structure) where the thickness of the electrolyte layer reaches its minimum.

2) The authors pay specific attention to the importance of the so-called bubble-free operation of the anode, while nothing is said about the cathode. From the Methods it is clear that the cathode was prepared in a conventional manner without using specific additives, such as, for example, PTFE employed during the anode preparation. Is the formation of bubbles selectively important only at the anode side? Will an improvement at the cathode side further improve the cell performance?

3) I found confusing the part where the authors discuss the changing of the ECSA of the anodes with and without PTFE. First, it is not clear to which material the measured ECSA corresponds. It is widely accepted that this term should solely be used when speaking about catalytically active sites. In this regard referring to the PTFE seems strange, as itself it should not catalyze the OER. Please correct this in both the main text as well as supplementary Fig. 13.

4) It is written that the long-term tests were performed under a constant stream of argon bubbling through the KOH electrolyte. Meanwhile, the authors demonstrate that at high current density the oxygen depolarization of the cathode does not occur. What would be the difference if the CFE cell will operate using an unpurged electrolyte?

5) Finally, I found several typos throughout the text, mainly in the supporting information. In particular, H₂O₂ in supplementary Fig. 3b and Ref. 37,39 appear in supplementary Table 5. The authors are encouraged to double-check the possible typos.

Apart from these issues, which should be answered, I must confess that the manuscript is well-written and organized and I enjoyed reading it.

RESPONSE TO REVIEWER COMMENTS

Title: A high-performance capillary-fed electrolysis (CFE) cell promises more cost-competitive renewable hydrogen
Authors: Aaron Hodges, Anh Linh Hoang, George Tsekouras, Klaudia Wagner, Chong-Yong Lee, Gerhard F. Swiegers, and Gordon G. Wallace
Manuscript number: NCOMMS-21-41100

***Kindly note that the original comments of the reviewers are reproduced after this response (for reference).**

Reviewer 1

We thank the reviewer for their complimentary and useful comments and respond as follows:

Major comments:

1. Regarding the reviewer's comments favoring the use of an energy efficiency based on the *lower heating value* (LHV) of hydrogen, instead of its *higher heating value* (HHV):

As noted in Supplementary Figure 1, we, respectfully, prefer the industry convention in which the energy efficiency of a water electrolysis cell is defined as:

*the net energy present in the hydrogen produced by the cell,
divided by the net energy consumed by the cell to produce it,
expressed as a percentage.*

The net energy present in hydrogen is its *higher heating value* (HHV), which equates to 39.4 kWh/kg hydrogen. This comprises of 33.3 kWh/kg of electrical energy (the *lower heating value* (LHV) of hydrogen) and 6.1 kWh/kg of heat energy. The minimum cell voltage at which an electrolysis cell can deliver the 39.4 kWh/kg of the HHV is the thermoneutral voltage (1.48 V at room temperature, 1.47 V at 80 °C). The minimum cell voltage needed to deliver the 33.3 kWh/kg of electrical energy of the LHV is the equilibrium voltage of water electrolysis (1.23 V at room temperature, 1.18 V at 80 °C). Thus, according to the above definition, an electrolysis cell producing hydrogen (with 100% Faradaic efficiency and zero crossover) at the thermoneutral voltage is 100% energy efficient.

We prefer the above definition because of its practicality and its applicability to hydrogen production. It accurately (and immediately) describes the percentage of the input energy that is captured and stored in the hydrogen produced. It also conveys the portion of the input energy that is wasted. For example, a 1 MW cell stack that is 75% energy efficient (HHV) can be immediately understood, even by non-experts, to direct 0.75 MW into hydrogen production and 0.25 MW into waste heat. The cooling requirement, namely 0.25 MW, is also directly indicated.

A person schooled in finance, rather than electrolysis, could, for example, readily discern that, if the above 1 MW cell stack were powered by renewable electricity costing \$10/MWh, then \$7.50 of each MWh would be used productively, to make hydrogen, and the remaining \$2.50 wasted. The cost of the cooling needed to remove the excess heat generated by the wasted \$2.50 would also be apparent; namely, 0.25 MW multiplied by the levelized cost of cooling.

Contrast this with the use of a definition for energy efficiency based on the *lower heating value* (LHV) as the reviewer suggests. The above 1 MW stack would then have an apparent energy efficiency of 62.3% (LHV). It would, nevertheless, still direct 75% of the 1 MW input (0.75 MW) into hydrogen production and 25% (0.25 MW) into waste heat.

In our experience, it is not easy for a non-expert to understand why a cell having 62.3% energy efficiency would turn 75% of its input energy into hydrogen (not 62.3%) and 25% into waste heat (not 37.7%). Indeed, it is generally difficult to explain the LHV convention to non-experts, including the many, diverse people involved in the hydrogen industry these days.

It seems to us that the reason it is difficult to explain is because the LHV convention is inappropriate to the *production* of hydrogen. It is, in our view, only appropriate to one specific *end-use* of hydrogen, after it has been produced, namely its use in a fuel cell. This is because fuel cells can only extract the electrical energy in hydrogen, which equates to the *lower heating value* of 33.3 kWh/kg hydrogen. However, green hydrogen will be used for many applications outside of fuel cells. Moreover, water electrolysis is about *making* hydrogen not *using* it.

We have read Lamy and Millet's paper on this matter (Journal of Power Sources, 2020, 447, 227350). They do not appear to us to conclude that practitioners should be using the LHV convention. Rather they state: "*Since both electricity and heat are required to electrolyze water, the most appropriate definitions of ϵ_{cell} are those which have homogeneous numerator and denominator expressions, i.e. which contain either the electricity and heat contributions together [the HHV convention], or only the electricity terms [the LHV convention]*".

They do caution that: "*Expressions of ϵ_{cell} that use the thermoneutral voltage at the numerator should be handled carefully ... If the expression used at the denominator contains only the electrical contribution, ϵ_{cell} values larger than 100% are obtained*" (which is a "thermodynamic nonsense").

We agree with that conclusion. Having studied water electrolysis cells operating endothermically, I can assure the reviewer that they cool rapidly, leading to quick declines in current (when the voltage is fixed) or increases in cell voltage (when the current is fixed). In common with many other physical processes, they give the illusion of being more than 100% energy efficient (HHV) only if they are examined over short periods under carefully selected conditions (i.e. where the heat they extract from the surroundings is not taken into account).

As noted above, hydrogen contains 33.3 kWh/kg of electrical energy and 6.1 kWh/kg of heat energy. An electrolysis cell operating endothermically, at a voltage below the thermoneutral voltage but above the equilibrium voltage, provides, in its applied cell potential, sufficient electrical energy *but insufficient heat energy to produce hydrogen*. The cell must and will extract heat from the surroundings to make up the 6.1 kWh/kg of heat energy needed in the hydrogen. If, at the applied cell potential, the missing heat energy cannot be obtained from the surroundings, then no hydrogen will be produced (i.e. the current will go to zero). For this reason, hydrogen can *never* be produced with more than 100% energy efficiency (HHV), even transiently. That is, at cell voltages below the thermoneutral voltage but above the equilibrium voltage, the cell may either operate with 100% energy efficiency (HHV) by extracting the missing but required heat from the surroundings, or, once that source of heat is exhausted, not produce hydrogen at all.

Our interpretation of Lamy and Millet's paper is that they showed that the HHV convention was often incorrectly formulated by failing to consider this fact. The conclusion that we draw from their paper is a need to properly formulate energy efficiency (HHV) and not a need to use the alternative LHV convention that is inappropriate to hydrogen production.

2. Regarding the reviewer's comments about "stable performance over extended periods", the following has been added to Supplementary Fig. 10 in the revised submission:
 - The current densities and voltages used have been included.
 - The change in voltage over the one-month study has been quantified and discussed.
3. Regarding the reviewer's comments about how we envision operating a scaled-up version of a capillary-fed cell stack at elevated temperature:
 - This is a little outside of the scope of the paper. However, as noted in Supplementary Tables 1 and 2, we believe that scaled-up capillary-fed cell stacks will be best operated under mildly exothermic conditions, such as at the cited 0.5 A cm^{-2} at 1.506 V. Under such conditions, the cells will be self-heating and capable of maintaining an operating temperature of $80 \text{ }^{\circ}\text{C}$ (i.e. they will not need to be placed in an oven). However, little excess heat will be generated, minimizing the cooling requirements.
 - Thus, Supplementary Table 2 shows that the cooling needed at $80 \text{ }^{\circ}\text{C}$ for a stack containing 500 capillary-fed cells, each having anodes/cathodes having 400 cm^2 geometric area covered by the catalysts, will be only 1.3 kW. This is well within the realm of air-cooling and radiative self-cooling may even be possible. Radiative self-cooling would involve the cell stack directly radiating the excess heat via, for example, externally fitted cooling fins. Air-cooling, if required, could potentially be achieved by slowly circulating a heat transfer fluid from around or through the stack (e.g. via cooling tubes in the bipolar plates) to a fan-cooled heat exchanger.
 - Comments to the above effects have been added to the text.
 - We agree that the fact that some portions of a capillary-fed cell stack are filled with gas while others are filled with liquid, may lead to localized heating and cooling effects within the stack, causing temperature gradients. Such gradients are not unusual nor fatal, with cell stacks containing separate gas- and liquid-filled volumes known and in use today (see, for example, US patents 8,999,135 and 11,005,117). Non-uniformities of this type are typically minimized by using computational flow dynamics when designing the cell stack.
4. Regarding the reviewer's comments about the calculations in Supplementary Section 1:
 - Details of these calculations have now been included in the section. The data was calculated using the excel sheet entitled "*Heat Output Calculator*", which has been provided as a Supplementary file.

Major minor comments:

5. The revised manuscript has been modified, as suggested by the reviewer, to:
 - Refer to the wt% of the electrolyte (rather than to the % only).

- Report the measured contact angle of the PES.
- Include the gas diffusion coefficients and gas solubilities for hydrogen and oxygen that were used to calculate the modelled curve for hydrogen gas crossover due to diffusion only (blue curve in Fig. 4f) – see references 35 and 36.
- Remove the word “very” from the sentence “*At current densities above 0.2 A cm⁻², hydrogen crossover trends very moderately upward*”. While there is, indeed, a significant deviation between calculated and measured hydrogen crossover at more than 0.2 A cm⁻², we were trying to say that, in absolute terms the deviation is small relative to crossovers measured in other systems.
- Clarify that the separator resistances in Table 1 include the resistance due to the electrolyte present within the separators. The technique described in Supplementary Fig. 5 does not subtract or correct for the ohmic drop of the electrolyte across the separator as suggested by the reviewer. Rather, it compares the resistance *between the electrodes of the conductivity meter* when that gap is filled only with electrolyte and when it is filled with electrolyte and the electrolyte-imbued separator. R_E is determined from the measured conductivity with electrolyte only. R_{S+E} is determined from the measured conductivity with electrolyte and separator. Full details are provided in the article that was referenced.
- Clarify that the anode electrode did not contain Sigracet carbon paper of the type used in the cathode, or any carbon whatsoever.

Minor minor comments:

6. The following corrections have been made as suggested by the reviewer:

- The sentence “...commit to *net-zero* carbon emissions by 2050 ...” has been re-stated to “... aim for *net-zero* carbon emissions by 2050 ...”
- References have been provided for the statements:
 - "The OPEX is, by far, the larger component of LCOH"
 - "... which recently culminated in asymmetric polymer electrolyte membrane (PEM) cells that directly produce one of the gases in a gas collection chamber..."
 - "with the entire liquid electrolyte typically replaced every 5-6 years."
- The sentence “*Any improvements in net energy efficiency create a similarly large decrease in the levelised cost of the produced hydrogen*” has been re-stated as “*Any improvements in net energy efficiency create a proportionally equivalent decrease in the levelised cost of the produced hydrogen*”
- The 8 and 4 in the denominator of Eqs. (1) and (5) have been moved to appear first
- The abbreviation “equ.” just above equation (5) has been replaced with the word “equation”

- The word “enhance” has been replaced by “enhanced” in the sentence "*Catalyst coatings from the resulting solution were found to display enhanced anode performance*"
- A clarification has been added to the effect that the electrodes and bipolar plates were pressed tightly together and against the separator by screwing in the anode and cathode positioning bolts shown in Supplementary Fig. 6.
- The sentence referring to the “geometric area of the electrocatalysts” has been changed to “the geometric area of the electrode that is covered with electrocatalyst”, as suggested.
- The resolution of the Figures has been improved, as depicted in the enlarged figures included separately in the submission. The figures are now each between 0.5 MB and 4 MB in size.
- The sentence referring to “a σ value of 0.75” has been corrected to refer to “a σ value of 0.75 mV.”
- The offending “therefore” has been removed from the sentence “While advective crossover is absent in PEM cells ...”
- The offending “the” has been removed from the sentence “The capillary-fed electrolysis cell also allows for a notably simplified the balance-of-plant”.
- The outcome of Supplementary Section 2 has been re-stated as 400 liters.
- Equations (6) and (7) have been retained in Supplementary Fig. 5 and removed from the paper text (to decrease the word-count of the manuscript).
- Comments and interpretation have been added to Supplementary Fig. 11, as requested.
- A statement to the effect that “the PTFE clearly increased the porosity of the electrocatalytic layer” has been added.

Additional comments provided to us on 16 December 2021:

7. Reviewer 1 later also provided the following statement

"Regarding the voltage improvements observed for bubble-free designs, have you considered the likely origin of dissolved gas on the equilibrium potential viz. Nernst equation? In our recent work we found dissolved oxygen and hydrogen to increase the open-circuit potential to well above 1.5 V in a zero-gap cell. When a gas-liquid interface is brought within diffusion distance to the electrode, this can strongly decrease the supersaturation, explaining the improvements found in bubble-free designs."

We agree with this statement by the reviewer. The text has been modified to the following:

The avoidance of bubble formation at $\leq 0.2 \text{ A cm}^{-2}$, which was likely due to the gas-liquid interface being within diffusion distance of the electrode, may also have decreased the supersaturation of the electrolyte, leading to a voltage decline. Elevated gas concentrations increase E^0 according to the Nernst equation. At higher current densities, supersaturation may have been needed at some electrode locations to produce the few bubbles observed.

Reviewer 2

We thank the reviewer for their complimentary and useful comments and respond as follows:

1. The selection of 27% KOH derived from our wish to utilize, as far as possible, the same electrolyte as conventional and historic industrial alkaline electrolyzers, which are most commonly stated in the scientific and patent literature to employ “6 M KOH”. As 6 M KOH corresponds to 26.7% KOH at 20 °C and 27.4% KOH at 80 °C, it was decided to use 27% KOH as an electrolyte best representing and most closely comparable to “6 M KOH” in the temperature range. We acknowledge that somewhat higher cell performance would have been achieved with 33% KOH, which displays the highest conductivity of any KOH solution at 80 °C.

Comments to this effect have been included in the text.

The molarity of the electrolyte does affect the rate of in-plane capillary-induced transport, because of the effects of viscosity, contact angle and surface tension. We confirm that we did not select 27% KOH based on its rate of capillary-induced transport. While we have not measured the transport rate of 33% KOH, we would not anticipate it to be significantly slower.

We initially thought that a KOH build-up may occur inside the separator as a result of the water consumption during electrolysis and sought to monitor the KOH molarity in the separator over time (which, unfortunately, proved impossibly difficult to do). The stable performance observed during, for example, the 1-month long water electrolysis experiment suggested, however, that no build-up of KOH occurs, at least in the present cell. We presume that water may also be osmotically induced to migrate from the reservoir up the separator to counteract any local increases in the KOH concentration. We acknowledge that the water- and ion-transport processes in the separator during electrolysis are not completely understood and need further investigation.

We never observed crystallization of KOH on any portion of, or inside the separator while it is dipped into a reservoir of 27% KOH. If the separator is filled with 27% KOH and then left on a bench to dry, crystals do start forming on the outer surfaces, but only after some days.

Comments to this effect have been included in the text.

2. In regard to bubble-free operation at the cathode: The graphs in Fig. 4c and Supplementary Fig. 9 represent the cell as a whole, including both the anode and cathode electrodes. Thus, both the anode and the cathode were largely bubble-free up to and including 0.2 A cm^{-2} , and substantially bubble-free above that to 1 A cm^{-2} . Accordingly, the cathode exhibited, in at least some significant measure, bubble-free performance.

The cathode comprised a Sigracet carbon paper gas diffusion layer (GDL) having two sub-layers: (a) a microporous carbon-PTFE layer at its front face and (b) a macroporous layer of PTFE-coated carbon fibres at its back face. A thin film of Pt/C catalyst was deposited on its microporous front face. We presume that, during operation, newly formed hydrogen migrated from the (wetted) catalyst layer through the micro-porous front face of the GDL into the (unwetted) macroporous layer at the back. Such a process may, conceivably, have been facilitated by the low surface energy and aerophilic nature of the PTFE in the microporous, and the macroporous sub-layers. Thus, while the cathode was *prepared* in a conventional manner as

noted by the reviewer, it was not *operated* in a conventional manner (i.e. with a fully flooded macroporous layer). The elements for spontaneous gas migration along aerophilic PTFE surfaces across the gas-liquid interface, leading to bubble-free operation, were therefore present in the cathode as well as the anode.

Comments to this effect have been included in the text and in the new Supplementary Section 3.

3. Regarding the effect of PTFE at the anode on the ECSA: The capacitance technique described in Supplementary Fig. 13 measured the double-layer capacitance of the anode, which is a function of the electrochemically active surface area (ECSA). The ECSA does, indeed, refer to the electro-catalytically active sites only and not to the PTFE surface area. We thank the reviewer for picking this up. It can be concluded that the presence of the PTFE increased the porosity of the catalyst layer and thereby also the area of electro-catalytically active sites. While the PTFE would, of course, be catalytically inert, it may have amplified the catalytic performance by facilitating migration of newly formed gas molecules along its highly aerophilic surfaces across the gas-liquid interface.

Comments to this effect have been included.

4. The reviewer is correct in noting that the long-term tests did not need to be conducted under a constant stream of bubbling argon (given that, at the current densities employed, we had earlier shown that oxygen depolarization of the cathode could not occur). We performed these tests under an inert atmosphere out of an abundance of caution and to be doubly certain that the observed performance was free of artefacts.
5. The typos noted have been corrected.

We trust that the above comments satisfactory responses to the reviewer's remarks.

Kindly note that other textual corrections, not requested by the reviewers, have been made to improve grammar and clarity.

ORIGINAL REVIEWER COMMENTS

Reviewer #1 (Remarks to the Author):

Review of the article "A high-performance capillary-fed electrolysis (CFE) cell promises more cost-competitive renewable hydrogen" by Aaron Hodges, Anh Linh Hoang, George Tsekouras, Klaudia Wagner, Chong-Yong Lee, Gerhard F. Swiegers, and Gordon G. Wallace.

This article details a very interesting novel configuration for water electrolysis, where the water is fed through the diaphragm allowing almost full bubble-free operation. The idea is elegant and very well described in the paper. Its various benefits are clearly explained and argued. In my opinion there is a lot of potential in the proposed approach, with benefits including a higher efficiency, simpler balance of plant, no shunt currents, and reduced gas cross-over. Some of the few downsides seem to be an electrode height limitation and some concern regarding whether heat can be removed effectively.

I can wholeheartedly recommend publication, as this constitutes important progress in the field. What follows are mostly minor comments, I hope the authors will consider, and a few more major suggestions. I would particularly like to hear argumentation, should the authors decide not to follow the first suggestion.

Congratulations with your excellent results and great paper.

Kind regards,
Willem Haverkort (j.w.haverkort@gmail.com)

PS: I may have a rudimentary understanding of where the voltage improvement of the bubble-free designs comes from, but hope you don't mind that I first further investigate and quantify the effect before I write it down properly.

major comments:

- An energy efficiency that can exceed 100 % does not make much sense. I would therefore be very much in favor of using the lower heating value of hydrogen as a reference, instead.

See also "Lamy, C., & Millet, P. (2020). A critical review on the definitions used to calculate the energy efficiency coefficients of water electrolysis cells working under near ambient temperature conditions. *Journal of Power Sources*, 447, 227350."

Eq. (21) of this reference, using the reversible potential of 1.23 V as a reference is a fine approximation. Insisting on using an efficiency based on the thermoneutral potential, the difference with the reversible potential should be added to the denominator, as in Eq. (20).

- "The capillary-fed electrolysis cell also demonstrated sustained stable performance over extended periods from 1 working day to 30 days continuously at 80 oC and room temperature, with periodic, manual, replenishment of the consumed water (Supplementary Fig. 10)."

Please provide the used current density (for the days figure) and voltage (for the hours figure).

There does seem to be a significant increase in voltage over the course of a month that seems to be swept under the rug. I would recommend to zoom in for the graphs of Supplementary Fig. 10 and comment on what could be the cause of this increase.

- "Instead, air-cooling or no cooling (i.e. radiative self-cooling of the stack) may be possible". It is not explained how the authors envision operating a scaled-up version at elevated electrolyte temperature without putting the electrolyzer into an oven. I envision that large temperature gradients will arise, with localized heating and little convective cooling. If instead operation very close to the thermoneutral voltage is envisioned there should be good insulation present?

- details for Supplementary Section 1 seem to be almost completely missing. Therefore I would recommend adding details on the performed calculations and the assumptions involved (or remove the section).

(The same holds true also for the LCOH calculations in Supplementary Figure 1, but there the calculation is arguable more straightforward).

major minor comments.

- use 27 wt% rather than % for clarity at several instances
- the value of the PES measured contact angle does not seem to be reported?

- "The solid line shows the gas crossover expected from diffusion only." please provide at least the assumed effective diffusivity and concentration difference.

- "At current densities above 0.2 A cm⁻², hydrogen crossover trends very moderately upward" The word "very" seems to downplay the very significant deviation (an order of magnitude difference between calculated and measured) too much and may be nuanced.

- If I understand correctly the "separator resistances" in table 1 are calculated by subtracting the ohmic drop of the electrolyte, which makes some sense, but may be non-standard? This resistance will still depend on the electrolyte conductivity, so is not a property of the separator. My impression is that most papers report the separator resistance including that due to the electrolyte (R_E), which makes more sense to me. Also, it is not made explicit how R_E is obtained, so there can still be some doubt over the exact procedure followed.

- it did not become fully clear to me whether the anode electrode contained the same Sigracet carbon paper as on the cathode.

minor minor comments:

- "to commit to net-zero carbon emissions by 2050 to limit global warming to 1.5 °C above pre-industrial levels" this 'commitment' is rather unbinding, and also the "to" is not a logical implication, so this sentence may be reconsidered.

-Please provide a reference for the statements

"The OPEX is, by far, the larger component of LCOH"

"which recently culminated in asymmetric polymer electrolyte membrane (PEM) cells that directly produce one of the gases in a gas collection chamber..."

and

"with the entire liquid electrolyte typically replaced every 5-6 years."

- "Any improvements in net energy efficiency create a similarly large decrease in the levelised cost of the produced hydrogen" similarly should be proportionally?

- The 8 and 4 in the denominator of Eqs. (1) and (2) should preferably appear first

- just above Eq. (5) it reads "equ." (2). Is this the journal preferred way of referring to an equation?

- "Catalyst coatings from the resulting solution were found to display enhance anode performance" enhance > enhanced

- "As the carbon paper could not be welded, the cathode was pressed tightly against its bipolar plate" How was this pressing done?

- "The current densities reported here are relative to the geometric area of the electrocatalysts." The use of the vague term "geometrical area" seems to leave still some room for confusion, especially when used in conjunction with "the electrocatalyst", perhaps change to "the geometrical area of the electrode that is covered with electrocatalyst"?

- The resolution of the figures is somewhat poor, especially the text. Perhaps consider using either vector drawings, or a higher resolution.

- "the capillary-fed cell displayed a σ value of 0.75, which was comparable to the bubbled cell at ~ 0.09 A cm⁻²" 0.75 should be 0.75 mV

- "The combined solubility and diffusion coefficients of hydrogen and oxygen are, however, 40-120-times higher in de-ionized water than in typical alkaline electrolytes at 80 °C³³⁻³⁵. While advective crossover is therefore absent in PEM cells" The word "therefore" seems a bit misplaced?

- "The capillary-fed electrolysis cell also allows for a notably simplified the balance-of-plant" remove "the"

- The outcome of Supplementary Section 2, 398 liter, seems to have a few too many significant digits for such a crude estimate.

- Eqs. (6) and (7) of Supplementary Fig. 5 are already in the paper and do not seem to fulfil a particular purpose here, so can best be removed?

- Some interpretation of Supplementary Fig. 11 seems desirable.
- Regarding Supplementary Fig. 12 one is left wondering why the PTFE increases the double layer capacitance. Any useful comments there?

Reviewer #2 (Remarks to the Author):

The manuscript “A high-performance capillary-fed electrolysis (CFE) cell promises more cost-competitive renewable hydrogen” introduces a novel concept of the water electrolysis systems, which demonstrates very high performance, even exceeding commercially available devices. The authors clearly describe a bipolar stack of CFE cells as well as their balance of plant. The results shown are well supported and should be of wide interest in the scientific community due to discovering of several important contributions that cumulatively led to a significant decrease in the cell resistance compared to the standard configuration of alkaline water electrolysis cells. While reported performance appears outstanding, I have several questions or comments which should be clarified to further improve the quality of the research.

- 1) The authors utilize 27% KOH electrolyte but never clarified their choice. It is expected that the molarity of the electrolyte will affect the in-plane capillary-induced transport of a liquid through a porous material. Besides, since the water will constantly be consumed during electrolysis, changing of the KOH molarity in a thin layer close to the electrode surface might be expected. Could authors comment on this? How they managed to avoid crystallization of KOH at the upper side of the separator (or inside its porous structure) where the thickness of the electrolyte layer reaches its minimum.
- 2) The authors pay specific attention to the importance of the so-called bubble-free operation of the anode, while nothing is said about the cathode. From the Methods it is clear that the cathode was prepared in a conventional manner without using specific additives, such as, for example, PTFE employed during the anode preparation. Is the formation of bubbles selectively important only at the anode side? Will an improvement at the cathode side further improve the cell performance?
- 3) I found confusing the part where the authors discuss the changing of the ECSA of the anodes with and without PTFE. First, it is not clear to which material the measured ECSA corresponds. It is widely accepted that this term should solely be used when speaking about catalytically active sites. In this regard referring to the PTFE seems strange, as itself it should not catalyze the OER. Please correct this in both the main text as well as supplementary Fig. 13.
- 4) It is written that the long-term tests were performed under a constant stream of argon bubbling through the KOH electrolyte. Meanwhile, the authors demonstrate that at high current density the oxygen depolarization of the cathode does not occur. What would be the difference if the CFE cell will operate using an unpurged electrolyte?
- 5) Finally, I found several typos throughout the text, mainly in the supporting information. In particular, H₂O₂ in supplementary Fig. 3b and Ref. 37,39 appear in supplementary Table 5. The authors are encouraged to double-check the possible typos.

Apart from these issues, which should be answered, I must confess that the manuscript is well-written and organized and I enjoyed reading it.

REVIEWERS' COMMENTS

Reviewer #1 (Remarks to the Author):

The authors have satisfactorily addressed my comments and suggestions and I have no further urgent recommendations (except perhaps to change the new Ref. 31 to the more relevant <https://doi.org/10.1016/j.jpowsour.2021.229864>). I repeat my enthusiasm for the promising bubble-free electrolyzer configuration and the thoroughness of the performed research and look forward to final publication.

Kind regards,

Willem Haverkort (j.w.haverkort@tudelft.nl)

Reviewer #2 (Remarks to the Author):

I would like to thank the authors for the proper revision of the manuscript and detailed answers to all the concerns. Congratulations for this excellent work!

RESPONSE TO REVIEWER COMMENTS

Title: A high-performance capillary-fed electrolysis (CFE) cell promises more cost-competitive renewable hydrogen
Authors: Aaron Hodges, Anh Linh Hoang, George Tsekouras, Klaudia Wagner, Chong-Yong Lee, Gerhard F. Swiegers, and Gordon G. Wallace
Manuscript number: NCOMMS-21-41100A

Reviewer 1

The new reference 31 has been changed as requested by the reviewer.

We thank the reviewer for their helpful comments.

Reviewer 2

As the reviewer requested no further changes, none have been made.

We thank the reviewer for their helpful comments.

Additional changes made

Kindly note that an author noticed an error in Supplementary Tables 1 and 2. The voltage and current densities used in these tables were drawn from Fig 3c, which was collected at 85 °C for the capillary-fed cell, not 80 °C, as had been stated in the text and captions of the Supplementary Tables. Accordingly, the operating temperature in Supplementary Discussion 1 and Supplementary Tables 1 and 2 has been changed to 85 °C, with very minor changes to the data in the last column of each table.

Minor grammatical improvements have also been made.